# Therapeutic Apheresis in Acute Relapsing Multiple Sclerosis: Current Evidence and Unmet Needs—A Systematic Review

**DOI:** 10.3390/jcm8101623

**Published:** 2019-10-04

**Authors:** Leoni Rolfes, Steffen Pfeuffer, Tobias Ruck, Nico Melzer, Marc Pawlitzki, Michael Heming, Marcus Brand, Heinz Wiendl, Sven G. Meuth

**Affiliations:** 1Department of Neurology with Institute of Translational Neurology, University Hospital Muenster, Albert-Schweitzer-Campus 1, 48149 Muenster, Germany; steffen.pfeuffer@ukmuenster.de (S.P.); tobias.ruck@ukmuenster.de (T.R.); nico.melzer@ukmuenster.de (N.M.); marc.pawlitzki@ukmuenster.de (M.P.); michaeloleg.heming@ukmuenster.de (M.H.); heinz.wiendl@ukmuenster.de (H.W.); sven.meuth@ukmuenster.de (S.G.M.); 2Department of Internal Medicine D, University Hospital Münster, Albert-Schweitzer-Campus 1, 48149 Muenster, Germany; marcus.brand@ukmuenster.de

**Keywords:** immunoadsorption, acute relapsing multiple sclerosis, plasma exchange, therapeutic apheresis

## Abstract

Multiple sclerosis (MS) is the most abundant inflammatory demyelinating disorder of the central nervous system. Despite recent advances in its long-term immunomodulatory treatment, MS patients still suffer from relapses, significantly contributing to disability accrual. In recent years, apheresis procedures such as therapeutic plasma exchange (TPE) and immunoadsorption (IA) have been recognized as two options for treating MS relapses, that do not respond to standard treatment with corticosteroids. TPE is already incorporated in most international guidelines, although evidence for its use resulted mostly from either case series or small unblinded and/or non-randomized trials. Data on IA are still sparse, but several studies indicate comparable efficacy between both apheresis procedures. This article gives an overview of the published evidence on TPE and IA in the treatment of acute relapses in MS. Further, we outline current evidence regarding individual outcome predictors, describe technical details of apheresis procedures, and discuss apheresis treatment in children and during pregnancy.

## 1. Introduction

In multiple sclerosis (MS), the complex interplay between environmental factors and susceptibility genes leads to the development of inflammatory brain lesions defined by oligodendrocyte death and axonal damage, recovery of function and structural repair, post-inflammatory gliosis, and neurodegeneration. Besides, disruption of the blood–brain barrier (BBB) and enhanced transendothelial migration of immune cells early in the course of MS likely contribute to the disturbance of neuronal integrity [1]. Numerous reports describe MS as a primarily T cell-mediated disorder [2,3]. However, findings such as immunoglobulin and complement deposits in demyelinating brain lesions, the presence of intrathecal immunoglobulin synthesis, and results from clinical trials on B cell depletion therapies suggest a pivotal role for B cells as well [4,5,6]. Several well-accepted roles for B cells in MS include the secretion of the central nervous system (CNS)-directed autoantibodies, B cell-dependent maturation of autoreactive CD4^+^ T cells, and dysregulation of cytokine responses [7,8,9].

Despite great advances in disease-modifying treatment, treatment for acute MS relapses has remained largely unaltered for the past 20 years, namely treatment with intravenous or oral corticosteroids [10]. The administration of high-dose intravenous methylprednisolone (IVMPS; up to 1000 mg daily) over a period of three to five days usually represents the first step in acute MS relapse treatment and has been endorsed by national and international guidelines, ever since a first prospective, randomized trial showed superiority of IVMPS compared to placebo [9,11]. The rationale here is primarily attributed to a non-genomic response, including a direct effect on cellular membranes, leading to a suppression of cell-mediated processes (suppression of immune-cell-migration, restitution of BBB integrity), as well as dose-dependent induction of T- cell apoptosis [12,13]. Although there is evidence for faster recovery of relapses by IVMPS treatment, there have been notable proponents for no effect of IVMPS treatment on long-term disability [14]. Furthermore, approximately one-quarter of the patient’s clinical improvement is not sufficient after the first course of IVMPS [15]. In this context, apart from extending steroid treatment to a higher dose (up to 2000 mg daily for five additional days), apheresis procedures, such as therapeutic plasma exchange (TPE) and immunoadsorption (IA) are considered as an alternative after their proven success in other neurological diseases [16,17,18]. Here, the lead (immuno)-pathogenetic principle for both apheresis modalities (TPE and IA) is based on the removal of circulating, pathogenic humoral factors such as autoantibodies, immune complexes and inflammatory cytokines, and the modification of pro-inflammatory mediators and co-stimulatory signals linked to T and B cell-mediated autoimmunity [19,20,21].

However, guidelines have no uniform recommendations on using TPE or IA in acute steroid-refractory MS relapses. Both procedures are recommended by the German Society of Neurology for escalation treatment of acute relapsing-remitting MS (RRMS) exacerbations not responding to the first IVMPS course [22]. The American Academy of Neurology also advises the use of TPE for adjunctive treatment of relapsing forms of MS (Level B), while IA is not addressed [23]. In 2016, the American Society for Apheresis published evidence-based guidelines for the clinical use of therapeutic apheresis, considering 16 neurological disorders. The modality recommended for most of these disorders was TPE. The clinical indication category assigns TPE for treatment of MS to category II (“apheresis accepted as second-line therapy”) and IA for treatment of RRMS to category III (“optimum role of apheresis therapy is not established”) [23].

Since several studies demonstrated that residual deficits persist after MS relapses and contribute to a stepwise progression of disability, fast and adequate therapy of relapses is indispensable, and the optimal treatment sequence has to be well defined [24,25]. Thus, we here aimed to conduct a review of the published literature that provides a general overview of available evidence using apheresis treatment in inflammatory demyelinating relapses, and in more detail outlines specific treatment-determining aspects.

## 2. Search Strategy

To conduct this review, we followed PRISMA (Preferred Reporting Items for Systematic Reviews and Meta-Analyses) guidelines [26]. The articles were independently screened for eligibility.

### 2.1. Inclusion Criteria

Studies had to meet the following inclusion criteria:Cohort: Reporting response rates of at least one patient with an established diagnosis of clinically-isolated syndrome (CIS) or RRMS (according to the McDonald or Poser criteria) or optic neuritis (ON) in absence of any other infectious or inflammatory disease of the CNS after treatment due to an acute relapse that is unresponsive to steroids [27,28,29,30].Treatment regimens: Apheresis was preceded by relapse therapy with high-dose steroids. No concomitant immunomodulatory relapse treatment was carried out simultaneously while using the apheresis procedures. Technical details of apheresis procedures adhered to current guidelines described in the 2016 consensus paper (number of TPE/IA courses: 4–7 (mean); patient’s estimated plasma volume (EPV) per session: 0.6–2.5; for TPE: Fluid replacement with albumin) [31].Outcome measure: Reporting at least one clinical outcome measure of apheresis; such as the expanded disability status scale (EDSS), its functional systems scores (FSS), or visual acuity (VA) [32].

Application of inclusion criteria: a. was mandatory for inclusion. If criteria b. and/or c. were not met, authors were contacted for additional data. Only in the case of insufficient information were the studies ultimately excluded.

### 2.2. Search Strategy

To identify studies, MEDLINE was searched for relevant articles published between 1 January 1980 and 30 April 2019 (in the English or German language). Additionally, we decided to evaluate abstracts from international conferences, namely Annual Meeting - American Academy of Neurology (AAN) and European/ Americas committee for treatment and research in multiple sclerosis (ECTRIMS/ ACTRIMS). Medical Subject Headings (MeSH) terms used were ‘multiple sclerosis’, and ‘plasma exchange’ or ‘immunoadsorption’. Furthermore, reference lists of published articles and abstracts were screened for additional studies (Figure 1).

## 3. Results

### 3.1. General Efficacy of Apheresis Procedures

#### 3.1.1. Therapeutic Plasma Exchange in Acute MS Relapse

TPE is an extracorporeal blood purification technique separating plasma from blood and involves the removal of patient plasma and the replacement with another fluid. TPE treatment of isolated ON, acute RRMS, and CIS, refractory to conventional pulsed IVMPS, has been evaluated in several studies [19,33,34,35,36,37,38,39,40,41,42]. Unfortunately, studies assessing functional outcomes of apheresis treatment used different outcome measures, usually referring to either the EDSS or its FSS. Reasoning that the most sensitive outcome was likely to be the one that addressed relapse-related disability, several studies primarily assessed changes in the ‘target neurologic deficit’ (TND), defined as the predominant symptom matching the FSS [19,34,35,38,42]. To simplify the statistical analysis, the majority of the reviewed studies classified recovery according to the response categories ‘no, mild, moderate, and marked’ (Table 1) [19,34,35,36,38,41,43,44,45]. Satisfying treatment success was mainly defined as ‘marked to moderate’ response. However, some studies also contained different classifications and/or other outcome parameters such as visual evoked potentials or gait/power scales from Weinshenker [33,39,42,46,47].

Here we outline both the response rates defined by the individual studies, as well as a uniform transformation of the data according to the ‘FSS-based relapse recovery model’ implemented by Conway and colleagues (Figure 2) [48]. The outcome model was based on the changes between the peak deficit and maximum recovery in FSS related to relapse (∆FSS = FSS relapse peak – FSS maximum recovery). Relapse specific changes in FSS were defined as a ‘good’, ‘average’, or ‘worse’ response to apheresis. Although patients’ individual EDSS or FSS were not outlined consistently throughout the different trials, we extracted the indicated FSS of ON, RRMS, and CIS patients where possible, taking all limitations of comparability into account. Table 2 gives an overview of published TPE data.

Studies

Up to now, only two prospective studies have been carried out [33,34]. In 1999, Weinshenker and colleagues conducted a randomized, sham-controlled and double-blind crossover trial, including 22 patients with acute, steroid-refractory relapses. While 13 patients were diagnosed with RRMS according to the Poser criteria, the other nine patients suffered from more or less specifically defined demyelinating disorders [30,33]. Half of the patients received TPE, while the others underwent a sham procedure in which the blood was separated within the apheresis system but recombined again prior to re-infusion. Patients that did not achieve a marked or moderate improvement crossed over to the other treatment arm. At discharge, 8 out of 19 patients (42%) had improved relevantly during the two-week course of active treatment, compared to 1 out of 17 patients (6%) in the sham group.

The second perspective but not the placebo-controlled study, published in 2009 by Trebst and colleagues, evaluated 20 patients with acute MS relapses and found a marked to moderate improvement in 18 out of the 20 patients (90%) following apheresis therapy [34].

Findings in the two prospective studies were underscored by several retrospective trials [19,35,36,38,39,40,41]. Two groups demonstrated comparable TPE response rates, revealing a marked to moderate response in 8 out of 11 patients (73%) and in 28 out of 35 patients (80%) at discharge [19,35]. Corroborating results from a small case series recorded a marked to moderate response in 3 out of 4 (71%) one month after TPE [36]. Contrastingly, Ehler and colleagues considered the response rate to be much lower: Following apheresis therapy, they detected a marked to moderate response in only 2 out of 11 patients (18%), while a mild response was observed in 8 out of 11 patients (73%) [38]. However, the remaining demographic characteristics and technical details of apheresis, including cycles and processed plasma volumes (PPV) did not differ substantially compared to other reviewed studies (Table 3). A slightly lower response rate was also demonstrated in a multicentre study conducted by Llufriu and colleagues with an improvement (not otherwise specified) in 11 out of 24 patients (46%) only [39]. As this study evaluated outcomes after six months, and thus later than the other trials, the sustainability of effects observed after TPE was questionable. Of note, Magana and colleagues also assessed the clinical course of 60 RRMS and CIS patients six months following TPE and revealed a rate of moderate or even marked response of 68% [41]. Several other studies with long-term data on apheresis-treated patients revealed sustained or even increased functionality during follow-up (1 to 14 months) with no new relapses occurring in between [19,34,35].

For a uniform transformation of data, according to the Conway model (Figure 2), six studies were accessible, outlining the patient’s FSS before apheresis and at discharge [19,34,35,36,38,42]. In total, 181 patients were included in this analysis. Forty-four patients (24%) experienced a good response, 89 patients (49%) an average response, and 48 patients (27%) a worse response.

#### 3.1.2. Immunoadsorption in Acute MS Relapse

IA has entered the field of MS treatment as a new important method for selective extracorporeal adsorption. In contrast to standard TPE, the eluted plasma can be reinfused during IA, suggesting a better tolerability and safety profile [50]. Two technical options have been developed: Single-use tryptophan-based adsorbers and reusable IgG Protein A containing staphylococci-based adsorbers (PrA-adsorbers). Single-pass devices are used for only one session, while their capacity is limited to approximately 2.0 to 2.5 L (L) of plasma volume. A treatment with tryptophan adsorber and conventional plasma separator results in an elimination of 30% of immunoglobulin IgG and IgM, 15% of IgA, 10% of the patient’s total protein and approximately 60%–70% of the fibrinogen during a single treatment, with a PPV of 2 L [51,52].

In contrast, a semi-selective (defined by the adsorption of mainly immunoglobulins (>95%), but not directed against a specific antibody) reusable PrA based devices can provide continuous IA to treat more than one EPV per session. Strong affinity for Fc fragments of immunoglobulins from any source is a remarkable characteristic of protein A. After one session with a PPV of 2,5 L a decrease in total serum IgG level of 87%, 55% of the IgA and 56% of the IgM level occur, without a clinically significant loss of fibrinogen (less than 15%) [53].

In the last two decades, six studies have been published on IA therapy for RRMS or CIS, including one prospective study [43,44,45,46,47,49]. All studies used tryptophan based adsorbers (Table 3). In order to compare these to the TPE studies, we separately reviewed the response rates for ON, RRMS, and CIS patients where possible and further assessed the patient’s FSS according to the Conway outcome assessment tool [48]. Table 4 gives an overview of the published IA data.

##### Studies

One prospective study has been conducted on IA in acute RRMS [46]. Koziolek and colleagues reported a significant improvement (defined as improved VA with 0.6 cc or more) in 8 out of 11 patients (73%) suffering from acute steroid-refractory ON (mean VA at baseline 0.12  ±  0.12 compared to 0.47  ±  0.32 at discharge). In four out of six patients (67%), visual evoked potentials could not be identified before treatment but were re-detectable after IA [46].

Various retrospective studies are in line with these data [43,44,45,47,49]. Mauch and colleagues reviewed the clinical course of 14 patients suffering from acute relapses of either RRMS (*n* = 12) or secondary-progressive multiple sclerosis (SPMS, *n* = 2). TND in 12 out of 14 patients (86%) significantly improved (assessed via EDSS, FSS not further specified) [47]. Corroborating results from Schimrigk and colleagues revealed 12 out of 15 patients (80%) with a marked to moderate response to IA [44]. In a subsequent retrospective multicenter study, Schimrigk and colleagues analyzed the largest cohort of MS patients treated with IA thus far, comprising six sites with 147 patients and 786 single IA treatments [45]. All patients suffered from an acute relapse of either RRMS (111 patients) or SPMS (36 patients). In 105 patients (71%), the affected TND improved functionally, including 88 patients (60%) with marked and 17 patients (11%) with moderate treatment response. Further studies indicated a marked to moderate response in 5 out of 10 patients (50%), and a marked to mild response (not precisely differentiated) in 53 out of 60 patients (88%) at discharge [43,49].

As the patient’s individual EDSS and FSS was not outlined consistently throughout the IA trials, a uniform transformation of data was not possible (only the data of 48 patients out of 3 individual studies were accessible) [43,44,47].

##### Comment

Guidelines on apheresis therapies currently refer to TPE only, since data on IA are considered less substantial [54]. Nevertheless, existing studies with individual outcome assessments indicated IA as effective with similar response rates compared to TPE (42%–90% for TPE vs. 50%–86% for IA) [19,33,34,35,36,38,43,44,45,46,47,49]. The limitations regarding the comparability of studies must be considered though. Criteria for patient selection and diagnosis significantly changed over time, and, therefore, characteristics of RRMS trial populations are diverging, probably resulting in lead-time bias [28,29]. In this context, a significant number of novel pharmacological agents have not just entered the field but even defined the treatment of active MS to date.

Additionally, the time points for apheresis therapy and evaluation of outcome parameters selected differed considerably throughout the trials. While EDSS assessment is universally familiar to MS clinicians and accepted by regulators, it has shortcomings in its variability between examiners, heavy emphasis on walking, and especially nonlinearity [55]. Consequently, several relapses associated with upper limb involvement are not reflected in terms of pre-existing gait impairment. Moreover, trials do not reflect whether a particular patient does not reconstitute at discharge or goes on to develop a persistent disability. In this context, in addition to evaluating the overall response, future studies should also cover the time between discharge and recovery, since complete but delayed recovery may still mean loss of independence and a need for rehabilitation or intermediate care. The main recovery has been suggested to take place within the first three months following relapse [24]. Hence, if an outcome is measured at discharge only, it may not be a suitable marker for overall improvement. On the other hand, a longer observation period carries the risk of detecting disabilities resulting from new relapses; confirmed disability progression at six months should be included as an outcome parameter in future study designs.

#### 3.1.3. Comparison of Apheresis Treatments (TPE vs. IA)

Despite the multitude of studies evaluating TPE or IA treatment of acute MS relapses separately, only a few studies compared both extracorporeal blood purification methods in terms of clinical efficacy, safety profile, and serological changes [21,54,56,57]. Assessing IA effectiveness is complicated even more by the co-existence of different IA systems (tryptophan based absorbers and PrA-based absorbers).

##### Studies

Two retrospective studies directly compared the efficacy of both apheresis treatments and described IA and TPE as equally effective for treating steroid-refractory relapses of MS [21,54]. Muhlhausen and colleagues included 140 patients with steroid-refractory exacerbation of MS and neuromyelitis optica (NMO), while Palm et al. compared the clinical efficacy of TPE and IA in acute relapses of RRMS as well as progressive forms, respectively [21,54].

In terms of treatment safety, IA is associated with fewer side effects and fewer contraindications [56,57]. Accordingly, the reviewed studies revealed a lower rate of adverse events (AE) during and following IA compared to TPE. Side effects and discomfort were apparent in 0.8%–15% of IA treatments [43,44,45,47,49]. In contrast, Koziolek and colleagues reported a higher side effect rate, with 22% mild and moderate AEs occurring during IA (12/55 IA treatments) [46]. Most AEs were classified according to the Common Terminology Criteria for Adverse Events (CTCAE) category I–II and were related to symptoms such as transient hypotension accompanied by dizziness and nausea, chest pain, palpitations, and headache. Side effects classified as CTCAE categories III–IV were mainly caused by vascular access and catheter-associated complications [45,46,47,49]. In this context, catheter dislocations (1/266 patients from all reviewed IA studies, specifying AEs in detail (Table 4)) and infections (3/266 IA patients), and catheter-associated thrombosis (6/266 patients) with one case of pulmonary embolism represent the most relevant risks after IA [45,46,47,49]. Moreover, one case of heparin-induced thrombocytopenia and one case of bradykinin-associated shock following a single administration of an ACE inhibitor have been described [49]. No study observed AEs classified as CTCAE category V following IA.

Notably, side effects reported for TPE were classified higher and were apparent in 24%–80% of all TPE treatments [19,33,34,35,36,38,39,42]. In particular, the risk of apheresis-related AEs with CTCAE categories I–III was higher for TPE than IA. Apart from the symptoms listed above, TPE AEs were also related to coagulopathy (including fibrinogen decrease), hypogammaglobulinamia, an allergic rash, paraesthesia, and anemia [33,35,38,39,42]. The rate of catheter-associated AEs were slightly higher compared to those of IA patients, with catheter dislocations (7/246 patients out of all reviewed TPE studies, specifying AEs in detail (Table 2), catheter-associated thrombosis (9/246 patients) and catheter-associated infections (11/246 patients), including one case of severe sepsis [19,33,34,35,36,38,39,42]. Additionally, one patient developed a catheter-associated phrenic nerve palsy, one patient a pneumothorax, and another patient experienced gastrointestinal bleeding, associated with hypofibrinogenemia [33,35,42]. Moreover, one patient died due to pulmonary embolism following TPE [33].

##### Comment

Based on the current literature, both treatments appear comparably effective. However, several studies described particular advantages in terms of safety for IA versus TPE [56,57]. IA avoids the removal of key plasma components with a milder impact on the cardiovascular system and a lower risk of allergic reactions and coagulopathy potentially underlying the more favorable safety outcomes.

According to American (ClinicalTrials.gov) and European (EudraCT) registries, further clinical comparative studies are in preparation. A prospective randomized trial, comparing both procedures in 60 RRMS and CIS patients with acute relapses, has been completed recently (NCT02671682); study results thus far unpublished.

In terms of technical options, all current IA studies employ single-use tryptophan-based absorber columns. Although PrA-based absorbers have been shown to be more effective in pure antibody depletion, the two techniques have not been compared in MS treatment thus far [58]. However, the responsiveness of steroid-refractory MS relapses to apheresis therapy is unlikely to depend on antibody clearance alone, since the clinical course of the disease is not correlated with autoantibody titers, and identification of crucial antibody targets remains elusive [59,60].

### 3.2. Individual Predictors of Apheresis Outcome

Comprehensive predictive models of individual treatment response to apheresis are lacking. This includes clinical, radiographic, and serological features to characterize individual differences in apheresis outcome. Several factors may exhibit a predictive value for apheresis response, including age, sex, monofocal or multifocal relapse manifestation, the affected functional system, previous success or failure to apheresis procedures in a preceding relapse, as well as technical aspects of TPE and IA [41,61]. However, the current literature is controversial, and whether predictors should be included in clinical decisions remains to be defined.

Using individual patient data from the respective studies, we performed an analysis of the literature using the Conway Matrix (Figure 2) to identify predictive factors for the TPE response. We included all studies that met the above inclusion criteria and outlined the complete sets of individual patient data, including all predictors we aimed to analyze. Overall, we included five studies with a total of 146 patients [34,35,36,42,62]. Statistical analysis was performed with R 3.6.1. The mean of the predictors was determined and categorized by the TPE response. Next, the data were scaled, centered, and depicted in a heatmap (Figure 3A). In order to determine the adjusted odds ratio of the predictors, a multivariate binomial logistic regression was performed (Figure 3B). Adjusted odds ratios with confidence intervals were displayed in a forest plot. Due to insufficient data in the IA arm, analysis in this cohort was not possible.

#### 3.2.1. Patient Dependent Variables (age, sex, neurological status, disease duration, and previous success or failure to apheresis)

Several studies have shown that the relapse recovery after IVMPS treatment linearly declines with age [48,63]. Correspondingly, our analysis of the literature revealed a positive association between poor TPE outcome and older age (Figure 3) [34,35,36,42,62]. This finding was statistically significant, as the adjusted OR for age was 0.93 (0.87–0.98, *p* <0.05).

Moreover, one observational study associated the male sex with a more favorable treatment response [61]. Of note, our analysis confirmed this observation, also indicating a trend towards a beneficial TPE response in male patients (Figure 3). However, this finding was not statistically significant.

Magana and colleagues further postulated a shorter disease course to be associated with a favorable TPE outcome [41]. We evaluated the impact of the disease course by implementing two groups: The first demyelinating event vs. not the first demyelinating event. Our analysis showed the superiority of the first demyelinating event compared to not, for treatment response to TPE and thus confirmed the above-mentioned observation (Figure 3) [34,35,36,42,62]. However, this finding was not statistically significant either.

Additionally, a retrospective study by Llufriu and colleagues revealed a positive correlation between the absence of disease modifying treatment (DMT) and more frequent response to TPE [39]. Since the respective demographic data of these patients are not mentioned in this study, it is not possible to specify whether this finding is based on a younger patient cohort or a shorter disease course. However, the predictive value of DMT has not been corroborated by other studies [41,62].

In addition, it can be assumed that a favorable response to apheresis in a preceding relapse provides a beneficial predictive value. In this context, rare individual case reports from the literature support this assumption [19,36]. However, the number of cases is small (*n* = 3) and not accessible for a systematic analysis.

##### Comment

To date, no patient-related response predictors have been confirmed for apheresis. However, regarding the discussed aspects above, our analysis revealed that older age, longer disease duration, and female sex were associated with poorer TPE outcomes (Figure 3) [34,35,36,42,62]. Although we performed uniform evaluation techniques and only included original studies that contained both the same patient cohort and a comparable TPE protocol according to current guidelines, we have to be aware of limitations concerning the validity of our findings, such as bias in the search strategy or methodological bias.

In the long-term, further clinical trials are needed to break new ground towards personalized MS relapse therapy. Apart from the factors mentioned, there are others of interest that have not been addressed in the studies thus far. These include the baseline annualized relapse rate, radiological, and immunological findings.

#### 3.2.2. Impact of the Affected Functional System

It has been discussed whether both subsequent long-term disability and post-relapse recovery are linked to the site of relapse manifestation [41,63,64,65]. Magana and colleagues described the highest response rates after TPE for brainstem and cerebellar impairments [41]. However, a relapse manifesting as ON has also been described to be associated with more favorable treatment response [48,65]. This is worth mentioning, as the proportion of patients suffering from acute ON varies considerably throughout the reviewed trials (TPE: 0%–100% [19,34,35,36,42,62]; IA: 25%–100% [43,44,45,46,47,49].

To provide better comparability, we analyzed the study data and extracted the indicated VA, transforming the values into a standardized outcome assessment tool as described above (Figure 2) [48]. Of note, demographic features such as age and sex, the dose of received steroids, the duration between symptom onset and apheresis treatment, and the number of treatment cycles did not differ between the trials.

We were able to extract a total of 151 patients from the 10 studies, treated with either TPE (*n* = 92) or IA (*n* = 59) due to acute steroid-refractory ON [19,34,35,38,42,43,44,45,47,49]. Based on the described outcome definition, 30 out of 151 patients (20%) showed “good” recovery, 66 out of 151 patients (44%) showed “average” recovery, and 55 out of 151 patients (36%) showed “worse” recovery at discharge.

IA resulted in a slightly higher proportion of patients with “good” recovery (14/59 patients, 24%) compared to TPE (16/92 patients, 17%). However, regarding the rate of non-responders, the available data indicated the inferiority of IA (31/59 patients, 52%) compared to TPE (24/92 patients, 26%).

In contrast, nine studies outlined the data of patients with relapse other than ON, including in a total of 119 patients [19,34,35,36,38,42,43,44,47]. As the respective studies did not further specify the affected functional system consistently, we included all these patients in one comparison between ON and relapses other than ON. Compared to the findings above, 29 out of 119 patients (24%) showed a good recovery (27/90 patients (30%) in the TPE, 2/29 patients (7%) in the IA group), 55 out of 119 (46%) an average recovery (40/90 patients (44%) in the TPE, 15/29 patients (52%) in the IA group), and 35 out of 119 (30%) a worse recovery (23/90 patients (26%) in the TPE, 12/29 patients (41%) in the IA group), at discharge.

Data on the comparison of ON vs. relapses other than ON regarding the TPE modality are separately outlined in Figure 3 (ON vs. No ON).

##### Comment

Except for acute ON, individual FSS was not indicated in the majority of the reviewed studies. This prevented further analysis of whether a specific disorder or symptom would respond more favorably to apheresis than another. Transforming VA to the new outcome model allowed us to analyze the response rate of ON to apheresis throughout the respective trials and compare them to relapses other than ON. Response rates to apheresis were homogeneously distributed, except for the study from Trebst and colleagues [19,34,35,38,42,43,44,45,47,49]. In this study, all five patients with ON showed a “worse” response to IA, and also during follow-up, only one patient improved [43]. Since both the technical data of apheresis and the demographic features of the patients did not differ from other trials, the cause for non-improvement remains elusive.

As already mentioned, we generated a heat map from the TPE data to demonstrate predictive factors of the TPE response (Figure 3). Although the finding was not statistically significant, there is a trend towards a favorable TPE outcome for relapses other than ON (no_ON, Figure 3). Unfortunately, we were not able to perform a corresponding analysis in the IA group due to missing data points. Nevertheless, findings encourage further investigations. Randomized controlled trials, investigating the response rates to both apheresis techniques regarding the relapse manifestation, with the treatment of the same individual plasma volume by TPE and IA, respectively, are needed.

#### 3.2.3. Time to Treatment Induction/Escalation

Time from symptom onset to apheresis initiation has been of interest in many studies [34,35,39,42,43,61]. Llufriu and colleagues demonstrated that time from relapse onset to TPE initiation was significantly shorter in patients who improved after treatment compared with those who did not respond [39]. When the days from symptom onset to apheresis initiation were stratified as ≤5 days, 16–60 days, and ≥60 days, a corresponding decrease of response rate was found (67% vs. 43% vs. 8%, respectively). Other groups also confirmed this time dependency [46,61]. Nonetheless, several studies demonstrated that even with a late TPE start patients markedly improved or recovered fully [34,38,49]. In this context, Trebst and colleagues showed treatment success with IA following a mean interval of 47.2 days—with patients being successfully treated even after more than 100 days following relapse onset [43].

Our analysis of the literature using the Conway outcome assessment tool revealed a slight, but statistically significant better TPE outcome, when initiated shortly after symptom onset (adjusted OR for shorter interval = 0.96 (0.92–0.99, *p* <0.05)) (Figure 3).

##### Comment

A short time interval between relapse onset and apheresis initiation is considered as a strong predictor of a good outcome, assuming nerve conduction in the early stages is blocked but not irreversibly damaged [19,35,39]. Consequently, it is recommended to initiate apheresis within six weeks after relapse onset [35,61]. However, apheresis should also be considered at later stages in patients with persistent symptoms. Unfortunately, no data exist thus far to aid the identification of potential responders. Nevertheless, we assume time to apheresis is one important modifying factor related to treatment outcome.

#### 3.2.4. Pre-Treatment with Steroids

Currently, there is no satisfying estimation of the contribution of steroid-treatment before apheresis to treatment success. The reviewed literature includes patients having received 2.0–15 g IVMPS prior to apheresis [19,33,34,36,38,43,45,46,47,49]. Steroids were administered after a median of 8–12 days (range: 1 to 68 days) after symptom onset and the time interval between the start of IVMPS and apheresis varied between 3 to 92 days. A limitation in all the studies is the absence of a placebo-treated control group. Thus, a spontaneous late improvement or a delayed effect of corticosteroids cannot be formally excluded. However, several aspects suggest the effects are directly associated with the apheresis procedure: No obvious correlation was apparent between the IVMPS dosage and treatment outcome, even when accounting for demographic and clinical features such as FSS, age, or sex [34,35,43]. Further, after a period of non-recovery prior to apheresis, symptoms began to improve soon after TPE or IA initiation [19,38,47,49]. In this context, several studies showed that most of the patients started to improve after the third cycle [19,38,47,49].

Further, we recently completed the first retrospective comparative study of double dose steroids and TPE in acute relapses of RRMS and CIS, including 145 patients [42]. All patients received 5g IVMPS before escalating treatment with either extended dose of IVMPS (2g/day over additionally five days), TPE (five cycles every other day, one EPV per session) or the combination of both (IVMPS, subsequently TPE). The treatment response was assessed according to the FSS-related outcome assessment tool described above (Figure 2). Good/average/worse recovery of relapse symptoms at discharge was observed in 60.9%/32.6%/6.5% of TPE patients vs. 15.2%/14.1%/70.7% of IVMPS patients. Of note, in patients who received TPE after escalated IVMPS treatment, the outcome was significantly inferior compared to those who received TPE as a first-line escalation treatment (multivariate odd’s ratio for worse recovery = 39.01 (10.41–146.18; *p* <0.001)).

##### Comment

Thus far, there is no standard treatment approach for ongoing relapse regarding steroids and apheresis modalities. Consequently, guidelines recommend both therapies as possible options for relapse escalation treatment but give no recommendations towards a specific treatment sequence [66].

Our retrospective comparison of escalated IVMPS vs. TPE revealed significantly higher response rates at discharge in the TPE group and in patients who underwent only one course of IVMPS prior to TPE [42]. As a possible explanation, it can be hypothesized that restoring BBB permeability with IVMPS treatment may, in fact, hinder the elimination of inflammatory mediators within the CNS via apheresis. Another explanation could be the longer time to apheresis treatment when conducted as the second instead of first escalation treatment.

Based on both assumptions, it may be preferable to choose a more aggressive therapy (TPE or IA) in cases of severe relapse, rather than the escalated IVPMS treatment. However, we recommend that the rapid admission of steroid-refractory patients to apheresis treatment without escalated IVMPS treatment should be prospectively evaluated.

#### 3.2.5. Number of Apheresis Courses and the Impact of the PPV

Guidelines currently recommend five to seven courses of apheresis treatment with a PPV according to 1.0 to 1.5 EPV [16]. An extension of further cycles can be considered on an individual basis in cases of non-response [22]. These recommendations are mostly derived from calculations of antibody concentrations. Antibody removal during a single TPE or IA treatment is limited, since (pathogenic) antibodies are often produced in abundance, with high tissue concentrations, and slowly equilibrated between their extravascular and intravascular distribution [67,68]. At least five separate treatments are required to eliminate 90% of the initial total antibody burden [69]. Further, especially for TPE, it is preferable to perform procedures every other day since exchange volumes per session are limited by the cumulative alteration of global hemostasis parameters and potentially increased bleeding risk [70,71]. Correspondingly, fibrinogen requires 48 h to recover to half of the pre-treatment level [70]. With regard to the studies discussed in this review, a comparable protocol was applied with regard to the technical aspects of therapeutic apheresis such as the volume of whole blood processed (number of cycles, number of plasma volumes exchanges per cycle), and replacement solution and vascular access, in line with recent national and international guidelines (Table 3) [31].

##### Comment

Current recommendations are based on calculations of antibody equilibration rates. However, there is plenty of evidence that apheresis treatment also eliminates additional soluble factors involved in acute inflammation [21]. Thus far, no studies exist examining the elimination kinetics of these additional factors. For clinical routine, the optimal number of cycles needs to be balanced in terms of achieving the best resolution of symptoms on the one hand, and keeping hospitalization times as short as possible on the other hand. In the reviewed studies, the first therapeutic effects typically occurred after a median of three apheresis cycles [19,38,47,49]. Consequently, based on kinetic considerations and clinical observations, a minimum of three to five cycles should be performed.

### 3.3. Apheresis Treatment in Special Situations

Although apheresis is an established therapy for adults, there is limited experience and literature on the application of apheresis in childhood or during pregnancy. The International Pediatric Multiple Sclerosis Study Group currently recommends TPE for children with severe relapses who do not improve after high-dose IVMPS or have contraindications; IA has not been implemented in the guideline [72]. However, established treatment standards for children are still lacking, and most apheresis protocols are derived from studies in adult patients [73].

Additionally, every fourth woman with MS experiences a relapse during pregnancy, and nearly every third suffers from a relapse in the first three months after birth [74]. Although the amounts of IVMPS in breastmilk are low, breastfeeding should be avoided for several hours after a high maternal dose and might occasionally cause temporary loss of milk supply [75]. Moreover, especially within the first trimester of pregnancy, high doses of IVMPS bear serious risks (preterm birth, a lower bodyweight of the child, and/or facial/palatal cleft) [76,77]. Thus, alternative treatment options are warranted. According to international guidelines, both apheresis procedures are recommended as an escalation or second-line therapy in pregnant patients with steroid unresponsive relapses [66,78].

#### Studies

In 2013, Koziolek and colleagues reported on the largest cohort of children undergoing apheresis treatment for severe attacks of demyelinating disorders (including RRMS; NMO-spectrum disorders and acute disseminated encephalomyelitis) refractory to IVMPS, demonstrating high and sustained recovery rates (88%) after five cycles of either TPE or IA [79]. These findings correspond to a retrospective study by Bigi and colleagues [80]. Five pediatric patients (median baseline EDSS 2.0; range: 0–3.5) were treated with TPE due to a severe attack of RRMS (median EDSS of relapse 6.5; range 4 to 7). The mean reduction in EDSS at discharge was 3 points, sustained at follow-up after three months.

Concerning treatment safety, De Silvestro and colleagues outlined a 5.6% rate of AEs in pediatric apheresis [81]. Corroborating rates were observed by Koziolek and colleagues, indicating four relevant side effects (hypotension, acute dyspnoea, catheter dislocation, and decreased serum fibrinogen) in 50 treatment courses in 4 patients (8%) [79]. However, in a retrospective study by Michon and colleagues, a much higher AE rate was detected for pediatric patients treated with either TPE or IA (55% of all procedures, *n* = 137) compared to adult patients (16.2% of all procedures, *n* = 86) [82].

Further, one retrospective study and several case reports suggest TPE and IA treatment for acute RRMS relapse during pregnancy and breastfeeding, considering them as rather a safe option [75,83,84]. In 2018, Hoffmann and colleagues conducted the largest retrospective study on this special cohort thus far, analyzing the use of tryptophan IA during pregnancy and breastfeeding in 24 patients. Twenty patients were treated with IA during pregnancy, and four patients received IA postnatal during the breastfeeding period. In 83% of patients, a rapid and marked improvement of the TND was achieved, defined either as an EDSS decrease of ≥1.0 point or improvement in VA ≥20% at discharge. Moreover, no clinically relevant side effects were reported in connection with the 138 IA treatments [75].

#### Comment

Published experience of apheresis procedures in children is limited, and most of the indications for treatment and the technical and procedural aspects of the procedure are also extrapolated from adult data and experience [82].

Koziolek and colleagues showed that apheresis in children suffering from acute relapses of demyelinating disorders seems to be as effective compared to treating adult patients [79]. However, special risk factors need consideration, including citrate toxicity, extravascular volume shifts, and difficulty related to vascular access [82]. Therefore, apheresis therapy in children requires a multidisciplinary approach involving expertise in children, intensive care medicine, and nephrology. Prospective studies are clearly warranted to optimize treatment protocols and to avoid permanent disabilities in this sensitive age group.

During pregnancy, apheresis represents a therapeutic option both in first and second-line relapse treatment. In this context, IA can be considered superior to TPE for several reasons. Plasma levels of numerous hormones undergo pronounced shifts during pregnancy [85]. The protective effect of pregnancy on MS disease activity seems to be at least in part, mediated by the immunomodulatory effects of these hormones. This supports the use of IA to preserve protective plasma proteins, instead of discarding them as with TPE. A further advantage of avoiding plasma substitution is the reduced risk of allergic reactions and infections, as well as the reduced impact of coagulation factors [86]. In contrast to TPE, most coagulation factors remain unaffected by IA, except fibrinogen, especially in terms of tryptophan based IA [87]. However, no fibrinogen substitution was required in pregnancy IA studies [75]. Thus, reduced bleeding risk in the perinatal period and reduced thrombotic risk due to the reduction of antithrombin can be achieved with IA treatment compared to TPE [88]. Moreover, since IA has a milder impact on the cardiovascular system, including hypotension, it can be assumed that it is superior in terms of placenta perfusion.

### 3.4. Closing Remarks and Outlook

Relapses are a hallmark of MS and often associated with significant functional impairment and decreased quality of life. Consequently, MS relapses need to be recognized and treated quickly using valid therapeutic methods. Although evidence for apheresis treatment in MS relapse is mostly derived from either case series or unblinded or retrospective cohorts—especially IA has thus far not been considered in international guidelines—both procedures have become an alternative to escalated and repeated pulsed steroid therapy. In respect to our recent comparison of escalated IVMPS and TPE, apheresis procedures should even be considered as a first-line escalation treatment [42].

In regard to the preferred apheresis modality and according to the literature, IA should be regarded as a method of equivalent therapeutic efficacy compared to TPE, probably offering greater safety in its use. This is particularly relevant in special populations such as children and during pregnancy.

As the long-term goal is to enable personalized MS relapse therapy, the predictive value for apheresis response is of high interest. In this regard, our analysis and several studies highlighted time to treatment as an important and modifiable factor related to outcome (Figure 3) [39,46,61]. Consequently, apheresis should be applied as early as possible in the course of a relapse. However, since some cases describe full recovery for patients even late after symptom onset, a strict regimentation to the suggested six-week period is not recommendable. Further, our analysis considers younger age, male sex, shorter MS disease duration, and a relapse not manifesting as ON to be associated with beneficial apheresis response (Figure 3). However, these parameters (eventually apart from sex) are associated with beneficial relapse outcomes in general [89,90]. Of note, using binomial regression models, both predictive values—male sex and shorter interval between apheresis initiation and symptom onset—were statistically significant (adjusted Odd-Ratio (OR) for age: 0.93 (0.87–0.98, *p* <0.05); OR for shorter interval = 0.96 (0.92–0.99, *p* <0.05). However, as in all systematic reviews, we have to deal with several limitations, such as bias from selection and publication of studies, availability of data, choice of relevant outcome, methods of analysis, and interpretation of heterogeneity. Therefore, we should keep in mind that systematic analyses should neither be a replacement for well-designed randomized studies.

In this regard, intervention studies with a prospective, randomized, controlled design are required to compare:(I)TPE vs. IA (with the same plasma volume exchange (TPE) or adsorbed (IA)), and(II)Escalated IVMPS vs. apheresis.

With regard to the study protocol, at least five cycles should be performed every other day. Considering the diversity of MS presentation, multiple patient-related clinical outcomes should be combined with surrogate endpoints [91]. Both outcome analyses at discharge and in the long-term (follow-up period of at least three months) are recommendable. Since it remains largely unclear by which cellular and molecular pathways apheresis affect disease activity, an analysis of blood samples and IA columns would also be of high interest.

There are several hypotheses regarding the mechanism of action of the different apheresis techniques, but their discussion is beyond the scope of this review. In this regard, previous studies provided evidence that the response to apheresis treatment is associated with the immunopathological pattern, indicating that the pattern of the two patients who showed signs of a humoral immune response benefited the most [40]. Anyways, knowledge about the predominant lesion pathology is usually unavailable in clinical routine as MS patients do not undergo a brain biopsy. Given the strong effects in the overall population of both TPE and IA, the procedures should be offered to all patients with severe and/or refractory relapses. We assume that, currently, the selection of the definitive technique will be determined by side effect profile and availability. However, trials are ongoing and will be published in the nearer future.

## Figures and Tables

**Figure 1 jcm-08-01623-f001:**
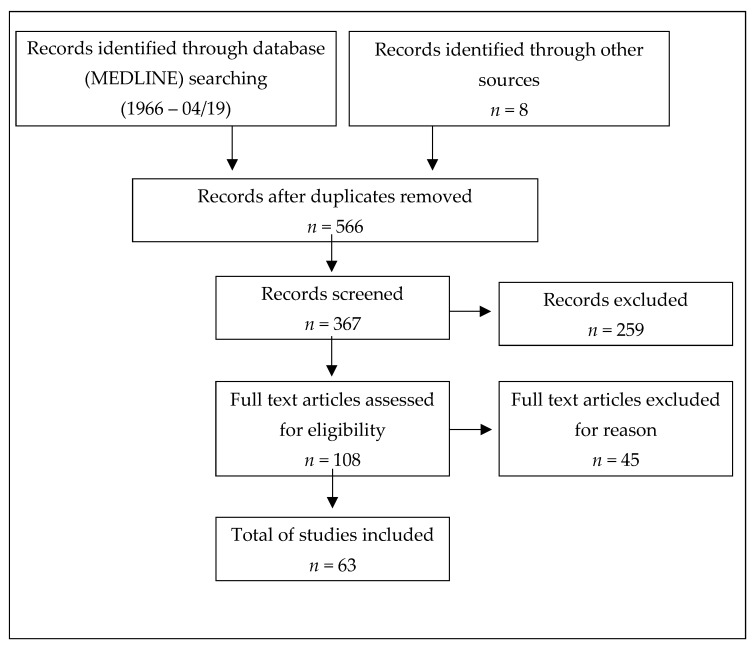
PRISMA flow diagram of the reviewed literature.

**Figure 2 jcm-08-01623-f002:**
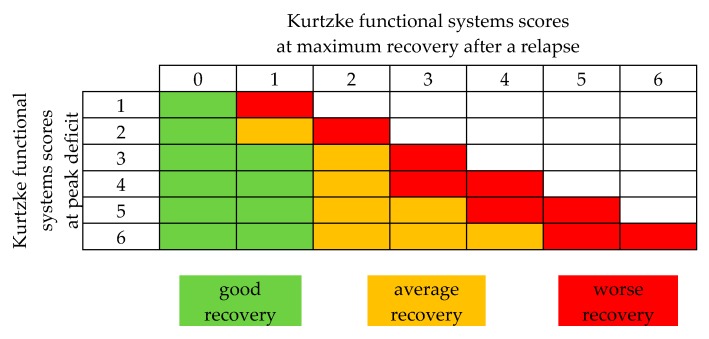
Functional systems score (FSS) based relapse recovery model. Good, average, and worse recovery is assigned based on the peak FSS, and the amount of final stabilized recovery FSS reached. Modified from Conway et al. [48].

**Figure 3 jcm-08-01623-f003:**
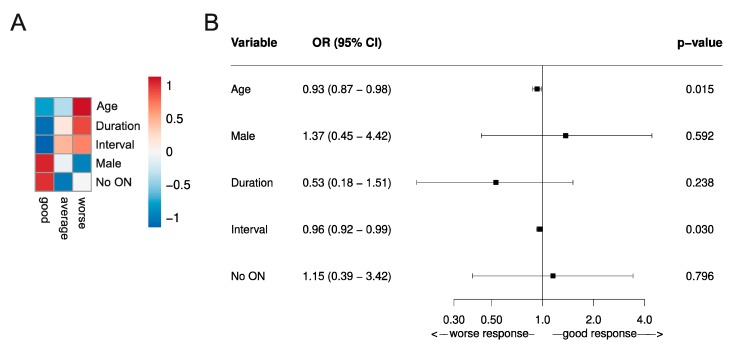
Predictive values of TPE response assessed by the functional systems score (FSS)-based relapse recovery model. (**A**) Predictive risk heatmap, applicable in individual patients, using three different outcome classifications (good, average, and worse) according to the Conway matrix (Figure 2) [48]. The heatmap depicts the scaled means of the variables. The red color indicates older patients (Age), with a higher prevalence of not suffering the first demyelinating event (Duration), a long interval between symptom onset and initiation of TPE (Interval), higher occurrence of male patients (Male), and a higher prevalence of relapse manifestations other than optic neuritis (no ON). (**B**) A forest plot of the predictive values of the TPE response in a multivariate logistic regression analysis. The x-axis represents the respective adjusted odd ratios for a worse versus good response. Odds ratios are outlined with a 95% confidence interval (CI; OR >1 no predictive value of treatment response, <1 statistic significant value associated with beneficial apheresis response). Data were generated in the RRMS and CIS patient cohort of five individual studies, including a total of 146 patients. All patients received TPE due to steroid unresponsive relapse [34,35,36,42,62].

**Table 1 jcm-08-01623-t001:** Classification of treatment response in the reviewed literature.

Level of Improvement	Definition
‘no response’	the same or even worse compared to baseline
‘mild response’	clinically detectable improvement but not relevant to function score
‘moderate response’	changes in function score
‘marked response’	major improvement or restitution of function

**Table 2 jcm-08-01623-t002:** Overview of publications on therapeutic plasma exchange in acute MS relapses.

Author	Publ. Year	Journal	Study Design	Disease Entity (no)	Sample Size	Outcome Parameter	Results in Regard to RRMS/CIS Patients	Results according to the Conway Matrix	Reference
Weinshenker BG.	1999	Ann Neurol	Prospective randomized, sham-controlled	RRMS (13), other IIDD (9)	22	EDSS (TND), gait/power scale	Relevant improvement in 8/19 patients (42%) after TPE vs. 1/17 (6%) after sham	N/A	[33]
Trebst C.	2009	Blood Purification	Prospective	RRMS (15), CIS (3), NMO (2)	20	EDSS (TND), VA, VEP	Marked to moderate improvement in 18/20 patients (90%)	Good response: 1/20 (5%) average response: 11/20 (55%), worse response: 8/20 (40%)	[34]
Schilling S.	2006	Nervenarzt	Retrospective	RRMS (6), CIS (5), NMO (2)	13	FSS, VA	Marked to moderate improvement in 8/11 patients (73%)	Good response: 1/11 (9%) average response: 4/11 (36%), worse response: 6/11 (55%)	[35]
Schroeder A.	2009	Aktuelle Neurologie	Retrospective	RRMS (22), CIS (9), SPMS (2), NMO (2)	35	EDSS (TND), VA	Improvement in 28/35 patients (80%)	Good response: 7/35 (20%) average response: 19/35 (54%), worse response: 9/35 (26%)	[19]
Habek M.	2010	Ther Apher Dial.	Retrospective	RRMS	4	EDSS	Marked to moderate improvement in 3/4 patients (75%)	Good response: 1/4 (25%) average response: 2/4 (50%), worse response: 1/4 (25%)	[36]
Ehler J.	2014	Ther Apher Dial.	Retrospective	CIS	11	EDSS (TND)	Marked improvement in 2/11 patients (18%), mild improvement in 8/11 patients (73%)	Good response: 1/11 (9%) average response: 7/11 (64%), worse response: 3/11 (27%)	[38]
Llufriu S.	2009	Neurology	Retrospective, multicentre	RRMS (22), CIS (5), other IIDD (17)	41	EDSS	Improvement in 11/24 patients (46%)	N/A	[39]
Magana SM.	2012	Arch Neurol.	Retrospective	RRMS (55), CIS (5) NMO (26), other IIDD (67)	153	EDSS (TND)	Marked to moderate improvement in 38/60 patients (63%)	N/A	[41]
Pfeuffer S.	2018	Multiple Sclerosis Journal	Retrospective	CIS, RRMS	99	EDSS (TND), FSS	N/A	Good response: 33/99 (33%) average response: 45/99 (46%), worse response: 21/99 (21%)	[42]

Abbreviations: CIS (clinically isolated syndrome), EDSS (Expanded Disability Status Scale), FSS (functional systems score), IIDD (idiopathic inflammatory demyelinating disorder), TND (target neurologic deficit), N/A (not applicable), NMO (neuromyelitis optica), RRMS (relapsing-remitting multiple sclerosis), SPMS (secondary-progressive multiple sclerosis), VA (visual acuity), VEP (visual evoked potentials).

**Table 3 jcm-08-01623-t003:** Overview of technical details on therapeutic plasma exchange (TPE) and immunoadsorption (IA )in acute MS relapse.

Author	IVMPS Refractory	Number of Cycles (range)	Possessed Plasma Volume	Replacement Fluid	Vascular Access	Reference
Therapeutic plasma exchange						
Weinshenker BG.	yes	7–14	1.1 EVP	5% albumin	CVA	[33]
Trebst C.	yes	3–7	3.0–4.2 L	5% albumin	CVA	[34]
Schilling S.	yes	4–6	3.0 L	4% albumin	CVA	[35]
Schroeder A.	yes	4–6	50 mL/KgBW	5% albumin	CVA	[19]
Habek M.	yes	5–10	1.5 L	5% albumin	N/A	[36]
Ehler J.	yes	3–8	2.2–3.5 L	5% albumin	PV, CVA	[38]
Llufriu S.	yes	5–15	125–166 EPV	5% albumin	N/A	[39]
Magana SM.	yes	2–20	1.1–1.4 EPV	5% albumin, FFP	N/A	[41]
Lammerding L.	yes	5	15 EPV	5% albumin	CVA	[42]
Immunoadsorption (tryptophan-based)						
Koziolek M.	yes	5	2.5 L	None	CVA	[46]
Mauch E.	yes	5–6	2.0 L	None	PV, CVA	[47]
Schimrigk S.	yes	3–6	2.0–2.5 L	None	CVA	[44]
Schimrigk S.	yes	3–6	2.0–2.5 L	None	CVA	[45]
Trebst C.	yes	5–7	2.5 L	None	CVA	[43]
Heigl F.	yes	6	2.0 L	None	PV, CVA	[49]

Abbreviations: CVA (central venous access), EPV (estimated plasma volume), FFP (fresh-frozen plasma), IVMPS (intravenous methylprednisolone), kgBW (kilogram per body weight), liter (L), PV (peripheral vein).

**Table 4 jcm-08-01623-t004:** Overview of publications on immunoadsorption in acute MS relapse.

Author	Publication Year	Journal	Study Design	Disease Entity (no)	Sample Size	Outcome Parameter	Results in Regard to RRMS/CIS	Results according to the Conway Matrix	Reference
Koziolek M.	2012	J. Neuro- Inflamm.	Prospective	RRMS, CIS	11	VA, VEP	Significant improvement in 8/11 patients (73%)	N/A	[46]
Mauch E.	2011	Nervenarzt	Retrospective	RRMS (11), SPMS (2); NMO (1)	14	EDSS (TND), VA	Significant improvement in 12/14 patients (86%)	Good response: 2/14 (14%) average response: 7/14 (50%), worse response: 5/14 (36%)	[47]
Schimrigk S.	2012	Aktuelle Neurologie	Retrospective, multicentr	RRMS (15), SPMS (9)	24	EDSS, VA	Marked to moderate improvement in 12/15 patients (80%)	Good response: 2/24 (8%) average response: 11/24 (46%), worse response: 11/24 (46%)	[44]
Schimrigk S.	2016	Eur Neurol.	Retrospective, multicentre	RRMS (111), SPMS (36)	147	EDSS	Marked to moderate improvement in 105/147 patients (71%)	N/A	[45]
Trebst C.	2012	Blood Purif	Retrospective	RRMS (8), CIS (2)	10	EDSS, VA, VEP	Marked to moderate improvement in 5/10 patients (50%)	Good response: 1/10 (10%) average response: 2/10 (20%), worse response: 7/10 (30%)	[43]
Heigl F.	2013	Athero- sclerosis supp.	Retrospective	RRMS	60	EDSS, VA	Marked to mild improvement in 53/60 patients (88%)	N/A	[49]

Abbreviations: CIS (clinically isolated syndrome), EDSS (Expanded Disability Status Scale), N/A (not applicable), NMO (neuromyelitis optica), ON (optic neuritis), RRMS (relapsing-remitting multiple sclerosis), SPMS (secondary-progressive multiple sclerosis), TND (target neurological deficit), VA (visual acuity), VEP (visual evoked potentials).

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
