# Peer review of "Therapeutic Apheresis in Acute Relapsing Multiple Sclerosis: Current Evidence and Unmet Needs—A Systematic Review"

_jcm, 2019, doi:10.3390/jcm8101623_

Round 1

Reviewer 1 Report

The publication is a review of the current state of literature on: therapeutic apheresis in acute relapsing multiple  sclerosis: current evidence and unmet needs. The manuscript is well prepared and solid. I feel that the paper has important contribution to the field. However, I have some concerns about the progress of research in this area. It should be noted that a significant part of the literature is not from recent years. I think it would be worth to supplement this knowledge from other medical databases.

Author Response

Dear Professor Andrès,

We would like to thank you and the reviewers for their careful reading and for the valuable comments and constructive suggestions to our manuscript ‘Therapeutic apheresis in acute relapsing multiple sclerosis: current evidence and unmet needs – a systematic review’(jcm-606950).

We hereby submit a revised version of our manuscript following the recommendations made by the reviewers. Additionally, please find a short rebuttal letter attached.

This version has been read and approved by all authors.

Yours sincerely,

Leoni Rolfes, MD

Comments from Reviewer 1:

It should be noted that a significant part of the literature is not from recent years. I think it would be worth to supplement this knowledge from other medical databases.

Response: Thank you for this helpful comment. Given that the evidence found on MEDLINE and correspondingly most of the represented studies, were published more than 10 years ago we decided to expand our search strategy. We also included abstracts having been presented at large international conferences, namely ‘Annual Meeting - American Academy of Neurology’ (AAN; abstract book published in Neurology) and ‘European committee for treatment and research in multiple sclerosis’ (ECTRIMS; abstracts published in Multiple Sclerosis Journal). Within the observation period indicated, this query did not result in any further studies meeting the inclusion criteria, apart from one study already included in the review (Lammerding et al. ‘Comparison of high-dose intravenous corticosteroids and therapeutic plasma exchange in acute relapsing multiple sclerosis’). Changes are outlined in page 3 line 117- 119 and page 4, figure 1. Please see the attachment.

Reviewer 2 Report

In this manuscript, Rolfes et al. summarized evidence on apheresis procedures in the treatment of acute MS relapses. Overall, the manuscript was constructed very well. However, it’s better to describe the author's conclusion more clearly in Closing remarks section. This would be helpful to understand the purpose of this manuscript.

There are a certain number of patients who are non-responders to standard corticosteroids treatment. Therefore, therapeutic approaches with apheresis including TPE and IA is important and will need very careful analysis when conferred to individuals with multiple sclerosis. This study is obviously needed, and summarized previous studies on apheresis procedures in the treatment of acute MS relapses. It is highly relevant to better understand beneficial effect of apheresis procedures. However, I want to make a cautionary note: I ask the authors to describe their conclusion more clearly in Closing remarks section. In addition, what is the limitation of apheresis procedures for therapeutic use without understanding cellular and molecular pathways of apheresis. These points would be helpful to understand advantages of apheresis procedures and the purpose of this manuscript.

Author Response

Dear Professor Andrès,

We would like to thank you and the reviewers for their careful reading and for the valuable comments and constructive suggestions to our manuscript ‘Therapeutic apheresis in acute relapsing multiple sclerosis: current evidence and unmet needs – a systematic review’(jcm-606950).

We hereby submit a revised version of our manuscript following the recommendations made by the reviewers. Additionally, please find a short rebuttal letter attached.

This version has been read and approved by all authors.

Yours sincerely,

Leoni Rolfes, MD

Comments from Reviewer 2:

I ask the authors to describe their conclusion more clearly in Closing remarks section. In addition, what is the limitation of apheresis procedures for therapeutic use without understanding cellular and molecular pathways of apheresis.

Response: Thank you for this helpful comment. We revised the discussion and closing remarks and discussed the impact of cellular and molecular pathways (Page 22-23). Please see the attachment.